# Characteristics of Women Seeking Cervical Cancer Cytology Screening in a Private Health Facility

**DOI:** 10.3390/medicina59091624

**Published:** 2023-09-08

**Authors:** Tizita Ashenafi, Clara Y. Stroetmann, Sefonias Getachew, Adamu Addissie, Eva J. Kantelhardt

**Affiliations:** 1Department of Preventive Medicine, School of Public Health, Addis Ababa University, Addis Ababa 366, Ethiopia; 2Global Health Working Group, Institute for Medical Epidemiology, Biometrics and Informatics (IMEBI), Martin-Luther-University Halle, 06112 Halle, Germany; 3Department of Gynecology, Martin-Luther-University Halle, 06120 Halle, Germany

**Keywords:** cervical cancer screening, cervical cancer screening uptake, private health sector, Ethiopia, self-initiation

## Abstract

*Background and Objectives*: Over 80% of cervical cancer cases in sub-Saharan Africa are detected at late stages, predominantly due to the lack or inaccessibility of prevention services. Public health facilities in Ethiopia offer free cervical cancer screening for eligible women. Besides the public health facilities, private providers also offer a variety of screening services at the patients’ expense. As the overall cervical cancer screening uptake in Ethiopia is still far below the 90% WHO target, coordination between all actors of the health system is key. This includes a close cooperation between the public and private sectors to combine the advantages of both to the benefit of all patients as well as media campaigns and community involvement to promote the self-initiation of screening. *Materials and Methods*: To gain insights into the utilization of cervical cancer screening in the private health sector, we conducted an institution-based cross-sectional study at Arsho medical laboratories in Addis Ababa. Every woman who came there for cervical cancer screening between 1 May and 30 June 2020 was asked to participate in a questionnaire-based, face-to-face interview about their socio-demographic background, cervical cancer screening experience and self-initiation of screening. A total of 274 women participated in the interviews. We further assessed the reproductive status of the patients, their risk factors for cervical cancer, source of information about the screening and barriers to cervical cancer screening. *Results*: The ages of the participants ranged between 20–49 years. The majority (over 70%) were married. A total of 37.6% reported self-initiating the screening. More than three-quarters of all interviewed women reported mostly using the private health care sector for all kinds of health services. *Conclusions*: While the Ethiopian government efforts on scaling up cervical cancer screening focus mainly on public health facilities, the private sector often does not get as much attention from policy makers. Efforts should be made to extend the government’s interest in cervical cancer screening and implementation research to the private healthcare sector.

## 1. Introduction

While cervical cancer has been declared a preventable disease targeted for elimination by the World Health Organization, it is still one of the most prevalent cancers in women worldwide [1,2]. Most cervical cancer cases, and related deaths, occur in low- and middle-income countries [2]. While screening and vaccination programs helped to drastically reduce the incidence of cervical cancer in Europe and North America, it is a major health threat for women in Sub-Saharan Africa—in some countries, it is the main cause of mortality by cancer in women [1,2]. Due to the lack or inaccessibility of prevention services, cervical cancer cases in sub-Saharan Africa are often detected at late stages, with lower survival rates and the need for aggressive treatment [1,2,3,4]. Without a massive increase in primary and secondary prevention, sub-Saharan countries are bound to face a growing burden of cervical cancer cases due to demographic and lifestyle changes within the population [1,2,3,4].

### 1.1. Cervical Cancer Screening Uptake in Ethiopia

The standard method for cervical cancer screening in Ethiopia is visual inspection with acetic acid (VIA), and the Ethiopian cervical cancer screening guideline aims for 80% of women to be screened at least once in their lifetime [5]. However, a systematic review in 2020 found that in Ethiopia only 13.46% of women aged 30–49 years had received at least one screening [6]. In order to upscale cervical cancer screening in the future, various improvements have to be made, including raising community awareness about cervical cancer and its prevention, training and equipping health care professionals to provide screening services, exploring alternative screening approaches and coordinating all measures on a national level [2,5,6,7].

### 1.2. Public vs. Private Health Facilities

In Ethiopia, cervical cancer screening is available in public as well as in private health facilities. There are major differences between both sectors: A. Public health facilities in Ethiopia offer cervical cancer screening free of cost [5], whereas private health facilities charge a fee that has to be covered by the patients themselves or their health insurance. B. Public health facilities often exclusively offer VIA, whereas women using the private sector can often choose between various screening methods (VIA, HPV-DNA-Testing, cytology: Pap smear or liquid-based) [5]. C. Especially due to the related costs and perceived benefits of the private sector, such as shorter waiting times, it can be assumed that the clientele of both sectors differs. Existing studies on cervical cancer screening often focus on public health facilities or it is not identifiable whether the included health facilities are public or private [6,7,8,9,10,11,12,13,14].

### 1.3. Objectives

This study was conducted in a private health facility to shed light on cervical cancer screening in the private sector as there are not sufficient data on the number of women using private cervical cancer screening services and their socio-demographic characteristics. Women might choose to receive screening in a private health facility due to the perceived benefits of the private health sector. What is more, women who are referred from the public to the private sector might receive care without being captured in the public monitoring and reporting system. To tackle cervical cancer in a nation-wide comprehensive approach, differences between the private and public health sectors should be identified and a strong collaboration formed.

The objective of this study is to assess cervical cancer screening experiences among women of reproductive age at Arsho medical laboratories (AML) in Addis Ababa, Ethiopia, to better understand the characteristics of women utilizing the private health sector for cervical cancer screening and to generate evidence for program and policy level action.

## 2. Materials and Methods

### 2.1. Study Setting

We conducted an institution-based cross-sectional study at AML. AML has two cytology-based collection sites in Addis Ababa. The geographical area served by AML includes five sub-cities (Arada, Addis Ketema, Bole, Kirkos and Yeka). The general services rendered at AML are clinical chemistry, pathology, histo- and cytopathology, microbiology, molecular diagnostics, serology and virology tests.

### 2.2. Data Collection

Two nurses interviewed all of the women who came to AML for a liquid-based cytology of the cervix between 1 May 2020 and 30 June 2020, apart from those women who came for a cervical cancer follow-up and those who did not agree to participate in the study. The consecutive sampling method was used to obtain the study population. The calculated sample size of 274 was reached, despite the patient flow being low, partly due to the COVID-19 pandemic. Informed written consent was obtained from every study participant.

The face-to-face interviews were based on a semi-structured questionnaire which had been developed after an extensive literature review. Primarily developed in English, the questionnaire was then translated to Amharic and back to check for consistency. The Amharic version was pretested in a comparable setting (at International Clinical Laboratories) on 5% of the calculated sample size, and a few questions were modified afterwards to ensure easy understanding. The final questionnaire consisted of five parts: A. socio-demographics, B. cervical cancer screening experience and self-initiation, C. reproductive status and risk factors for cervical cancer, D. information source and E. barriers to cervical cancer screening. The two nurses conducting the interviews had received a 1-day training by the principal investigator and were supervised daily during the data collection. The principal investigator also reviewed all records for plausibility.

### 2.3. Ethical Approval

Before the start of data collection, ethical approval was obtained from the Addis Ababa University School of Public Health and the Addis Ababa health bureau. Informed consent was secured from all the study participants.

### 2.4. Data Analysis

The collected data were entered into Epi data, cleaned, and checked for validity and completeness. Missing data are shown as “missing” in the tables. The reasons for missing data are that women declined to answer questions, were unsure how to answer correctly or were not asked. In some cases, obvious corrections were applied. RStudio was used for data analysis. Frequency and proportions were computed for a description of socio-demographic and other variables. Based on information about whether women self-initiated the screening or received a recommendation for screening by a health professional, we split our sample into two groups, and we want to describe the characteristics of both. We will refer to them as the “self-initiated” group and the “non-self-initiated” group.

## 3. Results

### 3.1. Socio-Demographic Characteristics of the Study Population

The “self-initiated” group consists of 103 women with a median age of 39 years; the “non-self-initiated” group consists of 171 women with a median age of 35 years (Table 1).

Patients aged 20–29 years constituted 10% of our self-initiated study sample, but 20% of the non-self-initiated group. Most participants in both groups were orthodox and most women were currently married. In both groups, the educational status as well as the monthly income of the participants were high, with a median income of 4500/3600 Ethiopian birr per month and less than 15% illiterate women.

### 3.2. Experience with Cervical Cancer Screening, Reproductive Status and Risk Factors

As Table 2 shows, most women at AML usually use private health facilities for their cervical cancer screening instead of public institutions. Most women who self-initiated the screening had their last screening within the last 2 years (56%), and 82% of the self-initiated group had been to any health facility within this time.

Reproductive status and selected risk factors for cervical cancer were also assessed—those are shown in Table 3. Less than 15% of women in both groups reported not having children.

Overall, 16.1% of women reported knowing someone with cervical cancer. Nearly half of the women who self-initiated the screening stated health facilities as their main source of information about cervical cancer. Of those who had used family planning, 75% (127 out of 169) reported use of oral contraceptives; 32 women reported taking OCP for 5 or more years. In both groups, about half of the participants stated that they have had only one sexual partner in their lifetime, while 95.2% reported three or fewer sexual partners. In total, 12.5% self-reported having experienced a sexually transmitted disease.

### 3.3. Barriers and Enablers to Uptake of Cervical Cancer Screening

Various barriers and enablers to screening uptake were assessed. In total, 44 women reported personally knowing someone with cervical cancer. Most women stated health professionals as their primary source of information on the issue (Table 4).

As services in private health facilities must usually be paid for, we assessed how women felt about the related costs. The cost for cervical cancer screening in AML was 450 ETH birr (about 13 USD) at the time of our study; most of our participants agreed that this price was either fair or cheap. As women might depend on their male partners’ financial and/or emotional support, we asked whether the patients had discussed the issue with their partner; 106 participants had done so and all of those received their partners’ supports. In contrast, from the women who had not discussed the issue with their partner, only 47% of women assumed that their partner supported their decision to have cervical cancer screening.

## 4. Discussion

Our study examined the socio-demographic composition of women using cervical cancer screening services in a private health facility and their screening experiences. The median age of our study participants was 36 years, and most had experienced a screening for cervical cancer before and preferred screening in a private health facility over a public facility.

### 4.1. Socio-Demographic Characteristics

We found that 47.8% of women using AML for cervical cancer screening mostly were in the primary target age group for screening (30–49 years), while 44 women were under 30 years old. According to the 2019 DHS survey, 69.3% of the urban female population in Ethiopia are under 30 [15]. A recent meta-analysis indicated an overall pooled prevalence of 13.46% for cervical cancer screening uptake in Ethiopia, reaching 18.38% in Addis Ababa. And, aligning with our results, it revealed a 4.58 times higher likelihood of screening for women aged 30–39 compared to those aged 21–29 [6], aligning with the higher risk of cervical cancer for women above 30 and the higher false-positive rates among the younger group [5]. Religious distribution mirrored the DHS survey, with Orthodox Christianity predominating [15]. Earlier research links religious beliefs to screening acceptability [10,16], warranting the inclusion of the variable in future study designs. Of our participants, 62.8% had completed secondary education or higher, compared to 27.4% of Addis Ababa’s population, according to the 2019 DHS [15], and 78% of our participants earned more than the 30,000 ETB/year that has shown increased screening uptake in an Ethiopian WHO steps study [12]. As women must pay for cervical cancer screening in private health facilities while it is cost-free in public institutions, we would expect primarily women who can afford the services to come to a private health facility. This seems to be the case, as most of our participants agreed that the price for screening was either fair or cheap.

In summary, our sample was older, more highly educated and wealthier than the average woman in Addis Ababa. This is not surprising, as several studies have shown that those factors can increase uptake of cervical cancer screening [6,12]. To discover possible differences between our sample and people in public health facilities, it is interesting to compare our sample to another sample of women who came for cervical cancer screening in Addis Ababa using Gahandi Memorial Hospital, a public health facility [16].

In our study, only 47.8% of participants were in the primary target group for cervical cancer screening (between 30 and 49), compared to 71% in Gahandi Memorial Hospital. The educational status of women seeking screening at Gahandi Memorial Hospital was higher than in our sample, with 80% having at least secondary education. The study in Gahandi Memorial Hospital did not report on the religious and financial background of their participants. In summary, our sample was older and less educated than the women seeking cervical cancer screening in Gahandi Memorial Hospital, Addis Ababa [16].

### 4.2. Reasons for Initiation of Cervical Cancer Screening

About one-third of our participants reported self-initiating the cervical cancer screening, which is similar to a facility-based study that was conducted in Addis Ababa in 2019 and found self-initiated cervical cancer screening to be 33.3% [17]. Health literacy, based on education and access to understandable health information, has been shown as one of the key factors for increasing the self-initiation of screening [6,8,9,10,11,12,13]. What is more, awareness is very important for self-initiated cervical screening. The women came by themselves because of the awareness they obtained from health professionals, the media, public health education and, in some cases, relatives. As most of our participants were married and all who discussed the screening with their partner received their partners’ support, this stresses the importance of involving men in awareness and communication programs and decreasing the taboos surrounding women’s health and cancer. A heightened awareness of cervical cancer prevention can contribute to increasing consciousness regarding cancers affecting this high-risk group, including vulvar, anal, and other forms of cancer [18,19].

The remaining two-thirds of the women participated in screening due to a recommendation by any health professional. Just like in a study by Bante et al. in northwestern Ethiopia, participants in our study reported health professionals as the most important source for information on cervical cancer [20]. This stresses the importance of the role of health professionals in educating patients about cervical cancer and its prevention—in this regard, the Ethiopian guideline for cervical cancer prevention advocates so-called Behavioral Change communication [7].

### 4.3. Experience with Cervical Cancer Screening, Reproductive Status and Risk Factors

In our study, 46.4% of participants used oral contraceptives and 12.8% reported having had any STD, compared to 36.4% oral contraceptive-users and 9.8% with a history of STD in a study from Mekelle, 2016 [11]. These discrepancies could potentially be attributed to differences in socio-demographic factors, access to healthcare, or awareness about contraception and sexual health between the population of both cities. The already-mentioned meta-analysis of Ethiopian studies found that women with multiple sexual partners as well as women with a history of STD were more likely to participate in screening [6]. Due to time and financial restrictions, STD testing could not be offered to our participants. About 27.7% of our participants had visited a health facility during the last year. The study from northwest Ethiopia that was mentioned earlier also found that visiting health institutions once or more per year increased the uptake of cervical cancer screening [20]. Status of HPV vaccination was not assessed in our sample, as vaccination programs in Ethiopia were only launched in 2018 and focused on 14-year-olds, whereas our sample was not in the target group.

### 4.4. Strengths and Limitations

Our study explicitly focuses on women receiving screening in a private health facility, which is a setting that has not received much attention in previous studies. By including all women who came for screening, we minimized selection bias. Initially, we aimed to assess factors in favor of self-initiation of cervical cancer screening, as we are aware of the importance of self-initiation for the national cervical cancer prevention strategy. However, as our study is lacking an appropriate comparison group, we decided to compare our results to the DHS survey as well as to a cervical cancer screening study conducted in a public hospital in Addis Ababa. As our study was conducted in Addis Ababa in an urban setting, this may not be generalizable to the wider population of the country. Due to the cross-sectional design of our study, it is not possible to show the temporal development of the assessed factors.

## 5. Conclusions

Our study sheds light on the characteristics of women using the private health sector for cervical cancer screening. Private and public health care services must form a strong collaboration to maximize the effectiveness of screening programs and monitoring and to assure good quality care in both sectors. In a next step, it would be interesting to explore patients’ perceived benefits and barriers to the private as well as the public health sector, to later generate interventions facilitating access to care for as many patients as possible and improving their experiences within the health system.

In general, it can be said that increasing the self-initiation of screening is an important approach to increase screening uptake. Community awareness should thus rise, possibly with the help of religious and community leaders. For health professionals, this means that they should be aware of the importance of counseling. Staff trainings should include explanations on how to convey understandable health information to the recipient. Soon, the structures built up for cervical cancer screening can be used for other health-related services such as breast examinations.

## Figures and Tables

**Table 1 medicina-59-01624-t001:** Socio-demographic characteristics among women of reproductive age using cervical cancer cytology screening at Arsho Medical Laboratories in Addis Ababa, Ethiopia, 2020.

		Women Who Self-Initiated the Screening		Women Who Were Recommended Screening by a Health Professional	
		*n* = 103		*n* = 171	
		Frequency	Proportion	Frequency	Proportion
Age	<30 years	10	10%	34	20%
	30–34 years	19	18%	44	26%
	35–39 years	25	24%	43	25%
	40–44 years	31	30%	32	19%
	45–49 years	18	17%	18	11%
	MEDIAN	39		35	
Religion	Catholic	3	3%	7	4%
	Muslim	13	13%	39	23%
	Orthodox	75	73%	111	65%
	Protestant	12	12%	14	8%
Marital	Married	71	69%	127	74%
	Widowed	14	14%	13	8%
	Unmarried	6	6%	18	11%
	Divorced	9	9%	10	6%
	Separated	3	3%	3	2%
Educational status	Illiterate	14	14%	21	12%
	Literate	4	4%	14	8%
	Primary education	15	15%	34	20%
	Secondary education	30	29%	41	24%
	Diploma and above	40	39%	61	36%
Monthly income	Low income (<61 USD ^1^)	12	12%	36	21%
	Middle income (61–194 USD ^1^)	67	65%	104	61%
	High income (>194 USD ^1^)	24	23%	31	18%
	MEDIAN in ETB	4500		3600	

^1^ Conversion rate 30 May 2020 1 USD = 34.156 ETB.

**Table 2 medicina-59-01624-t002:** Health-seeking behavior among women of reproductive age using cervical cancer cytology screening at Arsho Medical Laboratories in Addis Ababa, Ethiopia, 2020.

		Women Who Self-Initiated the Screening		Women Who Were Recommended Screening by a Health Professional	
		*n* = 103		*n* = 171	
		Frequency	Proportion	Frequency	Proportion
Most-visited health facility for CCS	Public	21	20%	39	23%
	Private	82	80%	128	75%
	Missing	0		4	2%
When was the most recent screening?	<1 year ago	18	17%		
	1–2 years ago	40	39%		
	2–4 years ago	25	24%		
	5 years ago	12	12%		
	>5 years ago	8	8%		
Last health facility visit?	<1 year	39	38%	37	22%
	1–2 years	45	44%	59	35%
	>5 years	19	18%	74	43%
	Missing	0		4	2%

**Table 3 medicina-59-01624-t003:** Reproductive status and risk factors among women of reproductive age using cervical cancer cytology screening at Arsho Medical Laboratories in Addis Ababa, Ethiopia, 2020.

		Women Who Self-Initiated the Screening		Women Who Were Recommended Screening by a Health Professional	
All		*n* = 103		*n* = 171	
		Frequency	Proportion	Frequency	Proportion
Number of children	0	12	12%	22	12%
	1–2	51	50%	62	36%
	3–4	25	24%	52	30%
	5–6	9	9%	14	8%
	7–8	4	4%	15	9%
	9–10	2	2%	6	4%
Age at first sexual intercourse	<16 years	6	6%	21	12%
	16–20	50	49%	93	54%
	21–25	31	30%	41	24%
	>25	15	15%	15	9%
	Missing	1	1%	1	1%
	MEDIAN	20		19	
Use of family planning	Yes	66	64%	103	60%
	No	37	36%	68	40%
Use of OCP	Yes	58	55%	70	41%
	No	19	19%	46	27%
	Missing	26	25%	54	3%
If OCP used, how long?	<5 years	43	74%	52	74%
	≥5 years	15	26%	17	24%
	Unknown	0	0%	1	1%
Lifetime sexual partner(s)	1	58	56%	98	57%
	2–3	42	41%	67	39%
	4–5	3	3%	4	2%
	6–7	0	0%	2	1%
History of STD	Yes	19	18%	16	9%
	No	84	82%	155	91%

**Table 4 medicina-59-01624-t004:** Cervical cancer screening barriers and enablers among women of reproductive age using cervical cancer cytology screening at Arsho Medical Laboratories in Addis Ababa, Ethiopia, 2020.

		Women Who Self-Initiated the Screening		Women Who Were Recommended Screening by a Health Professional	
		*n* = 103		*n* = 171	
		Frequency	Proportion	Frequency	Proportion
Know anyone with cervical cancer	Yes	18	17%	26	15%
	No	85	83%	145	85%
Information source	Health facilities	47	46%	46	27%
	Media	42	41%	88	51%
	Public health education	5	5%	16	9%
	Relatives	9	9%	21	12%
Service cost	Very expensive	5	5%	1	1%
	Expensive	19	18%	29	17%
	Fair	64	62%	120	70%
	Cheap	15	15%	21	12%
Transportation cost	Very expensive	3	3%	1	1%
	Expensive	24	23%	33	19%
	Fair	57	55%	110	64%
	Cheap	18	17%	24	14%
	Very cheap	1	1%	3	2%
Discussed screening with partner	Yes	44	43%	62	36%
	No	59	57%	109	64%
Support by the partner after discussion	Yes	44	100%	62	100%
Support by the partner without discussion	Yes	28	48%	51	47%
	No	31	53%	57	53%
	Unknown	0	0%	1	1%

## Data Availability

The data underlying the results presented in the study are available from the corresponding author. Data are not presented publicly for privacy reasons.

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
