# Peer review of "Characteristics of Women Seeking Cervical Cancer Cytology Screening in a Private Health Facility"

_medicina, 2023, doi:10.3390/medicina59091624_

Round 1

Reviewer 1 Report

Before any decision I recommend some modifications 

first is to rewrite better introduction to show more original output of such experiments 

second discussion section needs to be more attended including more references 

statistics needs much explanation and comparisons with similar literature findings 

bests

Quality of writing needs minor editing 

Author Response

Dear Editors and Reviewers,

We appreciate the time and effort you invested in reviewing our manuscript. Your comments and suggestions have been invaluable in refining our work, and we are pleased to present the revised version. To facilitate the understanding of the implemented changes, we have organized our responses in correspondence with the original review points:

  1. Introduction Enhancement:
    In response to your suggestion to "Rewrite better introduction to show more original output of such experiments", we have expanded the introduction and enhanced its overall structure.
  2. HPV Vaccination Clarification:
    Regarding the query about HPV vaccination exploration in the Results section, we acknowledge that our participants were not queried about their HPV vaccination status. This is attributed to the fact that the official vaccination initiative in Ethiopia was initiated in 2018 and targeted 14-year-old girls, a demographic that did not encompass our study cohort. We anticipate that forthcoming studies will exhibit heightened interest in this aspect as HPV vaccination rates rise. Please refer to the revised Paragraph 4.3.
  3. Self-Reported STDs Clarification:
    In response to the question concerning self-reported STDs in Paragraph 3.2, we confirm that STDs were indeed self-reported by participants. Due to resource and time limitations, we were unable to conduct STD tests on our study group. Kindly refer to the modified Paragraphs 3.2 and 4.3.
  4. Discussion Enrichment:
    We have addressed your recommendation to enhance the discussion section. This includes incorporating three new references that bolster the discussion's depth and relevance. One reference from Malawi, deemed irrelevant, has been excluded. We have updated citations of the Ethiopian demographic and health survey (DHS) data from 2016 to DHS 2019. Moreover, we have introduced a comparative analysis involving findings from a study conducted in a public hospital in Addis Ababa.
  5. Importance of Prevention Awareness:
    In relation to the query regarding the significance of prevention awareness on cervical screening uptake (Paragraph 4.2), we concur that awareness plays a pivotal role. We trust that the revised version of Paragraph 4.2 provides enhanced clarity on this aspect.
  6. Word Count:
    To align with the journal's guidelines, we have expanded the manuscript's word count.

We are grateful for your contributions, which have enhanced the quality and impact of our manuscript. Thank you sincerely for your time and consideration.

Best regards,

Clara Stroetmann, on behalf of all authors.
MLU Halle-Wittenberg
clarastroetmann@gmail.com

Reviewer 2 Report

Dear authors,

thank you for your work. A precious contribution in cervical screening in a developing nation. However, it provide moderately significant new data to the literature. 

Paragraph 3.Results section: HPV vaccination was explored? Since many international society provide HPV vaccines in Africa it would be interesting to know how many participants were vaccinated. To understand if it rains on already wet soil.

Paragraph 3.2: STD were self reported?

Paragraph 4.2: wad awareness of the importance of the moment of prevention in the visit an important factor for cervical screening uptake? (As Preti M, Selk A, Stockdale C, et al. Knowledge of Vulvar Anatomy and Self-examination in a Sample of Italian Women. J Low Genit Tract Dis. 2021;25(2):166-171. doi:10.1097/LGT.0000000000000585 demonstrated, the moment of cervical screening visit might be an important moment to integrate self examination and Preti M, Rosso S, Micheletti L, et al. Risk of HPV-related extra-cervical cancers in women treated for cervical intraepithelial neoplasia. BMC Cancer. 2020;20(1):972. doi:10.1186/s12885-020-07452-6 the importance of HPV effect on second primary tumors in this high risk group).

MInor

Author Response

(The authors gave the same response as above.)

Round 2

Reviewer 2 Report

I want to thank the authors for their precious work of revision.

Now the paper has more scientific soundness and well written introduction, and discussion.

Now the paper deserves publication.

Minor